# Rheological Behavior of the A356 Alloy in the Semisolid State at Low Shear Rates

**DOI:** 10.3390/ma16062280

**Published:** 2023-03-12

**Authors:** Oscar Martin-Raya, Sergi Menargues, Enric Martin, Maria Teresa Baile, Josep A. Picas

**Affiliations:** Department of Materials Science and Engineering, Universitat Politècnica de Catalunya (UPC), Rb. Exposició, 24, 08800 Vilanova i la Geltrú, Spain

**Keywords:** aluminum alloys, semisolid manufacturing, rheology

## Abstract

To control the semisolid processing of aluminum alloys produced by the additive manufacturing technique, an exhaustive knowledge of their rheological behavior is required. In the semisolid state, metallic materials can show rheological characteristics similar to those of polymers, so semisolid state shaping is one of the currently considered routes for additive manufacturing with metallic materials. In this work, an approximation of the rheological control of the A356 aluminum alloy for its subsequent 3D manufacturing was carried out at a very low shear rate. A continuous cooling rheometer was designed and used, evaluating the influence of different process parameters on the viscosity variation of the aluminum alloy in the semisolid state. The results show an anomalous flow variation, indicating dilatant, and not thixotropic behavior, for very low shear rates.

## 1. Introduction

The ASTM 52900:2015 standard is an attempt to regularize a new field of manufacturing processes, Additive Manufacturing (AM), which has generated a lot of interest in diverse engineering sectors and has great potential for expansion. This standard classifies the AM processes in the following categories: (1) Binder Jetting, (2) Directed Energy Deposition (DED), (3) Extrusion, (4) Material Jetting (5) Powder Bed Fusion (PBF), (6) Sheet Lamination and (7) VAT Photopolymerisation [1]. These technologies open new manufacturing possibilities regarding the production of complex geometries, impossible to be produced by any other means, and the multi-functionality of the components. Despite the short time which has passed since the development of these technologies, their viability has been proven for different materials. Nowadays, the efforts are focused on the proper control of the processing parameters to assure the quality and properties of the printed components. Exploiting the whole potential of these technologies will require the development of new materials, and/or modifications of the existing ones, to optimize their rheological behavior during printing. In general, the above-described technologies are the most extended techniques for the manufacturing of metallic parts by additive manufacturing, but all of them have high costs associated to the heating systems, i.e., electron beam or laser.

This work has the objective of being able to develop processes to print thixotropic aluminum alloys based on the Fused Deposition Melting (FDM) techniques, extensively used for the printing of thermoplastic components, which could significantly reduce the costs associated with metal printing. These techniques involve additive manufacturing of extruded material: the material is forced to pass through a nozzle by applying pressure, it is deposited at a constant speed and it solidifies on top of the material previously deposited. This idea is difficult to implement in metal printing because the melting point of most commercial alloys is much higher that the melting point of polymers. For this reason, it is necessary the production of metallic parts through the adaptation of the material extrusion technologies, is based on the control of the rheological properties of the materials in a semisolid state (SSM). The semisolid state can be obtained for certain alloys in the temperature range where solid and liquid phases coexist, in which the material shows a specific rheological behavior that determines its formability and exhibits special improved rheological properties which help control viscosity and benefit laminar flow [2]. However, a very strict control of flow of the material through the nozzle is necessary to avoid obstruction of the nozzle [3]. Nozzle clogging is one of the most significant process errors in current fused filament fabrication 3D printers, and it affects the quality of the prototyped parts in terms of geometry tolerance, surface roughness, and mechanical properties [4].

The authors base this study on their extensive knowledge about the thixotropic forming of aluminum alloys [5,6,7,8] whose rheological behavior in the semisolid state will be key for its printing by FDM type technologies.

To achieve the objectives of this work, it is essential to know the rheological behavior of the materials formed by SSM and determine the most appropriate rheology. Viscosity is the most important parameter of rheology as it provides information about its fluidity and, consequently, its capacity to flow through the nozzle [9]. In some materials, the viscosity decreases when fluid is agitated. This property is known as thixotropy and semisolid manufacturing takes advantage of it because the lower the viscosity, the lower the effort required to deform the fluid and fill the mold [8] or flow through a nozzle in an additive manufacturing process.

The viscosity η of a fluid can be defined as the resistance of a material to deformation as the strain rate increases [10], and quantitatively, the ratio between the shear stress applied to deform the fluid τ expressed in Pa and the strain rate $\dot{\gamma }$ expressed in s^−1^ [11]. The term viscosity is often used synonymously with apparent viscosity [10].

Viscosity can be predicted with Equation (1) [12]:(1)$$\eta =A\;\exp\left(B\cdot f_{s}\right)$$
where *A* is related to the pre-exponential viscosity, *B* to the activation energy, and *f_s_* is the solid fraction. Accordingly, Equation (1) can be considered as a form of the expression of the Arrhenius equation [13]:(2)ηT=ηoexp ER·T
where *E* is the activation energy for viscous flow, *η_o_* is the pre-exponential viscosity, *T* is the temperature in *K* and *R* is the gas constant (8.3144 J·mol^−1^·K^−1^).

This power law behavior can also be analyzed using the Ostwald-de-Waele model [14], given by Equation (3):(3)$$\eta =m\cdot {\dot{\gamma }}^{n-1}$$
where *m* is the flow consistency index (Pa·sn), and *n* is the flow behavior index (dimensionless). Power-law fluids can be subdivided into three different types of fluids based on the value of their flow behavior index. For *n* < 1 pseudoplastic fluid, *n* = 1 Newtonian fluid, *n* > 1 dilatant fluid (or shear-thickening fluid).

There is a very important dispersion (around 400%) in the viscosity values in the literature for aluminum alloys, depending on the process used to evaluate them [13]. It has also been noted that, depending on the publication, the viscosity has a difference of 10^3^ factor. Ilda and Guthrie discuss this property in order of mPa·s after being cross-referenced for seven investigations [13], on the other hand, Barman and Dutta discuss it in order of Pa·s [15] evaluating the rheological performance of the A356 aluminum alloy in the semisolid state as a function of shear rate and cooling rate

Parameters as solid fraction or previous deformation processes, can lead to very different values [12,14]. Viscosity values around zero were found for low values of solid fraction, which was justified because liquid metals can exhibit Newtonian behavior [14].

In accordance with previous results, the objective of this study is to carry out a rheological control of the A356 aluminum alloy, for its subsequent 3D manufacturing, at a very low shear rate.

## 2. Materials and Methods

In this work a continuous cooling rheometer was designed and used, evaluating the influence of different process parameters on the variation of the dynamic viscosity of the aluminum alloy in the semisolid state. Viscosity will be calculated using a rotational viscometer, consisting of a coaxial cylinder sensor system that compares assigned torque/shear stress with the measured strain/strain rate (Figure 1). For this, specific software has been designed that relates these magnitudes with the final viscosity.

According to Schram, the formulas for shear stress, *τ* (Equation (4)), and shear rate, $\dot{\gamma }$ (Equation (5)), are [11]:(4)$$\tau =\frac{T}{2\cdot \pi \cdot L\cdot R_{i}^{2}\cdot Cl}$$
where *T* is the torque [N·m], *L* is the rotor length in contact with the melted alloy [m], *R_i_* is the radius of the rotor [m] and *Cl* is the torque correction factor that considers internal friction forces of the machine and has been taken by running a previous test without a melted alloy.
(5)$$\dot{\gamma }=\frac{2\cdot \omega \cdot R_{a}^{2}}{R_{a}^{2}-R_{i}^{2}}$$
where *ω* is the angular velocity [s^−1^], *R_a_* the radius of the cup [m] and *R_i_* the radius of the rotor [m].

Viscosity can be obtained by dividing the shear stress and the shear rate, and combining Equations (4) and (5), the viscosity will be obtained from Equation (6):(6)$$\eta =\frac{\tau }{\dot{\gamma }}=\frac{T\cdot \left(R_{a}^{2}-R_{i}^{2}\right)}{4\cdot \pi \cdot \omega \cdot L\cdot R_{a}^{2}\cdot R_{i}^{2}\cdot Cl}$$

In accordance with the exposed principles, a rheometer has been designed and built for this experiment (Figure 2). The rotor is a 50 mm diameter Cu-Cr-Zr CW106C bar (*R_i_*), with internal air-cooling control, and the cup is a silicon carbide 110 mm inner diameter crucible (*R_a_*).

The dimensions of the rotor and cup were designed to maintain a small temperature gradient, throughout the alloy volume, during the experiments. Simulations with PRroCAST 10.0 software (ESI Group, Rungis, France) were taken proving that this condition was assured (Figure 3).

A K-type thermocouple was used to measure the temperature of the aluminum alloy throughout the process. Corresponding solid fraction, *f_s_*, present in the slurry was calculated based on the Scheil equation (Equation (7)) [16]:(7)fs=1−TM−TTM−TL11−kp
where *T_M_* is the melting temperature of pure aluminum, *T_L_* is the liquidus temperature at initial composition of the studied alloy and *k_p_* the partition coefficient of the alloy.

The experiment was undertaken with the hypoeutectic A356 aluminum alloy, with 7% of silicon and 0.3% of magnesium (Table 1).

The A356 aluminum alloy is commonly used in thixocasting and rheocasting processes due to its wide gap between solidus and liquidus temperatures, good pourability and weldability, high specific strength and good corrosion resistance. Consequently, this alloy is commonly used in the fabrication of components for the aeronautical and automotive sectors.

The SSM process of the A356 aluminum alloy must be carried out in a temperature range in which the alloy is in a two-phase region (liquid + solid phase of α-aluminum) with a desirable morphology and solid fraction. In this sense, it is considered basic to achieve a globular solid phase morphology. In addition, for semisolid 3D manufacturing, the liquid fraction percentage must guarantee a continuous extrusion flow to avoid clogging in the extrusion nozzle, which could imply the discontinuity of the printing process. However, the use of excessively high percentages of liquid fractions can decrease the viscosity and may favor the formation of liquid drops at the exit of the extrusion nozzle, which will alter the fluidity of the semisolid slurry and consequently a correct printing process. On the contrary, the use of low percentages of liquid fraction (high solid fraction) will involve high extrusion forces, which makes the process tangled or eventually unfeasible. The solidification of the A356 aluminum alloy starts at 613 °C [17]. The temperature range between liquidus (613 °C) and solidus (557 °C) allows the possibility of carrying out a controlled semisolid process.

For each experiment, a quantity of 1.3 kg of A356 aluminum alloy was melted up to 620 °C by means of an induction furnace. Once the set temperature was reached, the bar agitated the melted alloy in a pre-set velocity while cooling to 592 °C, in order to assess the early stages of solidification (Figure 4).

To avoid thermal shock between the copper bar and the fluid, it was necessary to heat the bar to a temperature of 350 °C when starting the rotation. The agitation time was no longer than 20 s. The semisolid slurry was obtained when the alloy contained a certain percentage of solid and liquid fraction. The management of the cooling rate and the viscosity during the solidification is necessary for a correct formation of the semisolid slurry.

In additive manufacturing processes, shear rates are technically low, while most studies of viscosities for this material have been carried out for relatively high shear rates. Therefore, in this work it is proposed to carry out experiments for very low strain rates, from 10.5 s^−1^ (100 rpm) to 15.7 s^−1^ (150 rpm) and 18.3 s^−1^ (175 rpm).

The designed and built rheometer was validated using the rheology charts provided by the Anton Paar Company (Ashland, VA, USA) for the same aluminum alloy.

## 3. Results and Discussion

During the material preparation step, the dendritic microstructure of the A356 aluminum alloy was broken-up by the mechanical stirring and solidification control of the SSM slurry that allowed us to obtain samples with a globular microstructure. Figure 5 shows the as-cast microstructure of the A356 aluminum alloy after conventional solidification and the alpha globulized microstructure of the solidified SSM slurry.

The evolution of viscosity with the temperature of the experiments carried out with continuous cooling, and at a constant lower strain rate, from 10.5 s^−1^ to 18.3 s^−1^, is shown in Figure 6. Considering the direct relationship between the solid fraction of the A356 aluminum alloy and temperature, the evolution of viscosity with the solid fraction has also been plotted on this graph.

The evolution of the viscosity with the shear rate showed an anomalous behavior, contrary to most of the studies carried out [15,16,18,19]. For the low shear rates used in this study, the viscosity level increased with increasing shear rates.

From Figure 6 it can be seen that for the lower value of shear rate (10.5 s^−1^), and for values under 19% of solid fraction, no appreciable difference in apparent viscosity was observed. When solid fraction was between 19% and 30%, under low stirring velocity, it was observed that viscosity raised, and when the solid fraction exceeded 29%, it was observed that viscosity increased rapidly. This rapid increase would be due to increasing structural bonds between particles with the increase in the solid fraction [15]. The viscosity showed from the beginning an evident exponential dependence on temperature.

The regression analysis of the values shown in Figure 6, allows us to determine the values of A and B of Equation (1). The calculated values are shown in Table 2 and Figure 7 shows the evolution of viscosity as a function of solid fraction. Table 2 and Table 3 also show the standard error of estimated parameters “A” and “B”, “*η_o_*” and “E” and the square of the correlation (R^2^), which is a useful value in linear regression. This value represents the fraction of the variation in one variable that may be explained by the other variable (a value of one indicates a perfect correlation).

The regression analysis of the above graph allows us to determine the values of $\eta_{o}$ and E of Equation (2), as shown in Table 3. Bringing together Equations (1) and (2), A can be interpreted as the pre-exponential viscosity, *η_o_*, and B is related to the energy of activation, E, which is lower when the shear rate is increased.

The power law regression analysis of the results reveals a value of n greater than one. So is, the fluid shows shear-thickening behavior [14]. The viscosity rises with the increase of the shear rate giving a dilating behavior, or shear-thickening behavior, and not of a thixotropic one [20], and it is consistent with the value of n previously exposed. The general behavior is justified basically to the morphological changes in the solid particle agglomeration [21]. However, at low shear rates, post-isostructure processes may correspond to rapid agglomeration followed by slow neck growth between spheroids and thickening [22]. There is conflicting evidence in the literature as to whether behavior during shear rate transitions, for non-dendritic semisolid metals, are shear thinning or shear thickening [23]. Solid particles suspended in the liquid have a tendency to agglomerate, which increases the viscosity of the semisolid slurry, increasing this tendency with the application of external shear forces [3] especially at low shear rates [24].

On the other hand, at small deformation strain, the predominant mechanism is sliding between solid particles, and the plastic deformation mechanisms of solid particles is less than that produced at higher strain rates [25], and the incorporation of solid particles into the liquid flow becomes more important (Figure 8). Simultaneously, at low shear rates, solid particles can aggregate into groups capable of forming a rigid network, or percolating network, which leads to an increase in the viscosity of the semisolid slurry [26], the breakdown of these groups or clusters took more time and need more shear stress [27]. It would be necessary to reach a critical shear rate to reverse the process, a situation in which the material would behave more normally.

In all cases, viscosity will increase with decreasing temperature (increase in solid fraction). This increase is quite fast from 0.25 *f_s_* as a consequence of structural bonds increase between the solid particles with the increase of the solid fraction [15].

Given that liquid segregation can occur at high forming speeds in semisolid extrusion [28], in a subsequent additive manufacturing process, the deposition of the material in the semisolid state would have to be carried out at relatively low shear rates, and this dilatant phenomenon could affect its deposition, which could even lead to the clogging of the nozzle [29].

## 4. Conclusions

The present work reports a study on the rheological behavior of the semisolid A356 aluminum alloy slurry. Based on the results, the following conclusions can be drawn:(1)A mechanical stirring and solidification control system was manufactured, which allowed the material to be worked in a semisolid state, resulting in a globulized alpha microstructure of the solidified A356 aluminum alloy.(2)The rheological behavior of the A356 aluminum alloy was performed a very low shear rates: from 10.5 s^−1^ to 18.3 s^−1^. The experimental results showed that the alloy behavior was shear-thickening, that is, viscosity increased with increasing shear rate.(3)This anomalous behavior is interpreted by the tendency of the solid α-aluminum particles, suspended in the liquid phase, to agglomerate and the subsequent formation of groups or clusters for low shear rate values. These clusters increase the viscosity of the aluminum alloy in semisolid state.(4)In all cases, the viscosity increased with decreasing temperature. When the solid fraction exceeded 29%, it was observed that viscosity increased rapidly.(5)After applying mechanical stirring in semisolid state, aluminum ingots can be used in a subsequent additive manufacturing process, based on the Fused Deposition Melting (FDM) techniques of metallic materials (3D thixo-printing).(6)The globulized alpha microstructure of aluminium ingots is considered beneficial, as a suitable microstructure to not impair the continuity of the semisolid alloy through the nozzle in the thixo-printing process. In this process, the material must be reheated to the semisolid state, in which the coexistence of a globular solid phase surrounded by a liquid phase, will allow the extrusion of the semisolid slurry with the appropriate viscosity. However, for very low shear rates, problems of possible clogging of the nozzles of additive manufacturing printers are foreseen, due to the dilating nature of the alloy.(7)The printing of metallic materials in semisolid state will introduce manufacturing possibilities never before foreseen. The reduction in the cost of printing will expand the use of metal 3D thixo-printing to diverse industrial sectors.

## Figures and Tables

**Figure 1 materials-16-02280-f001:**
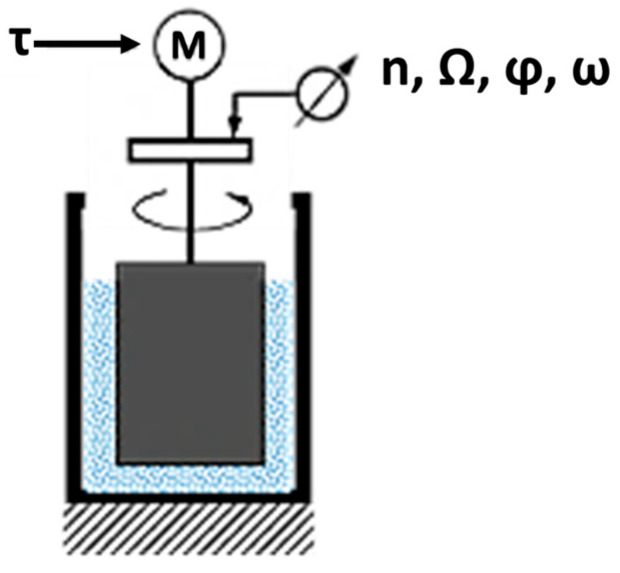
Controlled Stress (CS)-Rheometer, adapted from [11].

**Figure 2 materials-16-02280-f002:**
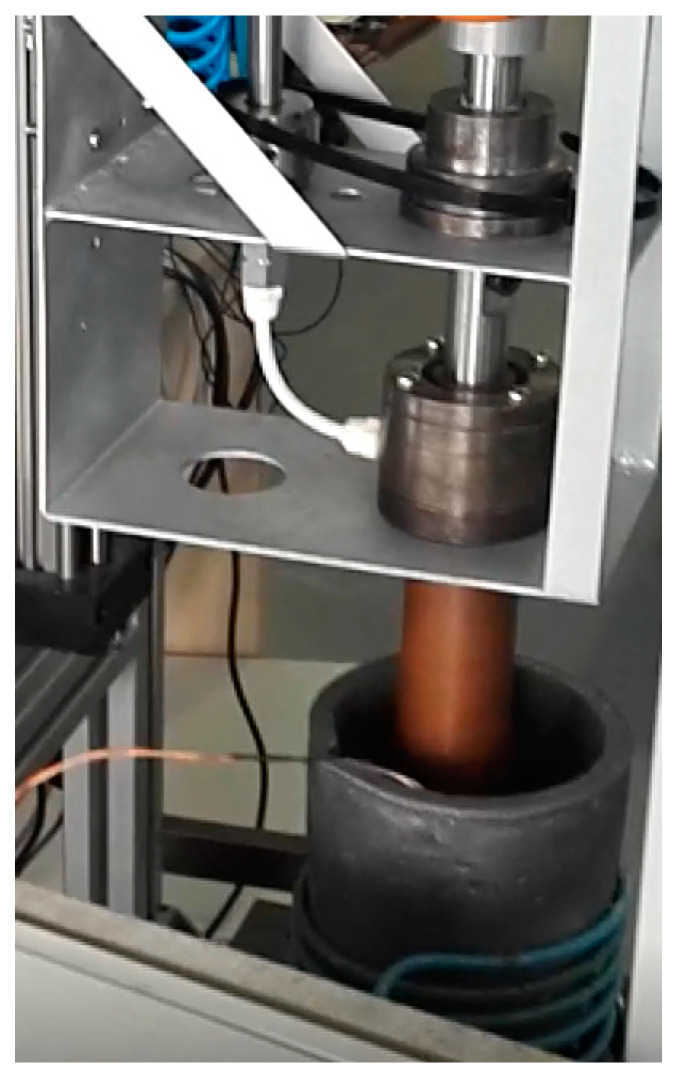
Experimental Controlled Stress (CS) -Rheometer.

**Figure 3 materials-16-02280-f003:**
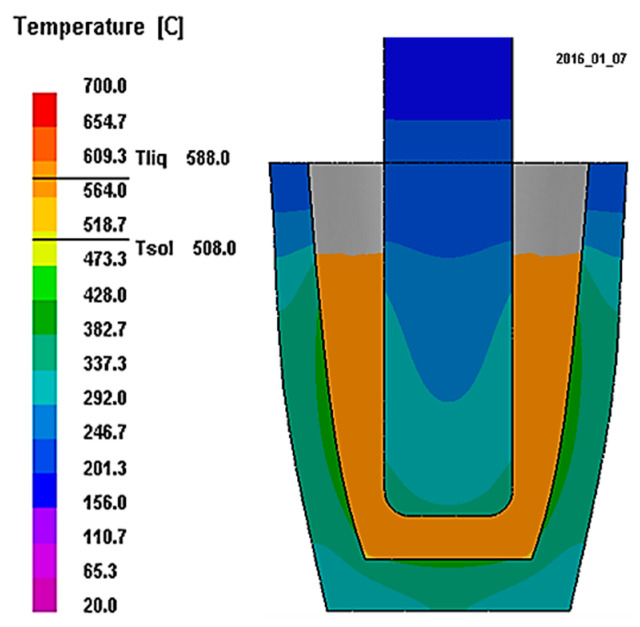
Cooling simulation with ProCAST software.

**Figure 4 materials-16-02280-f004:**
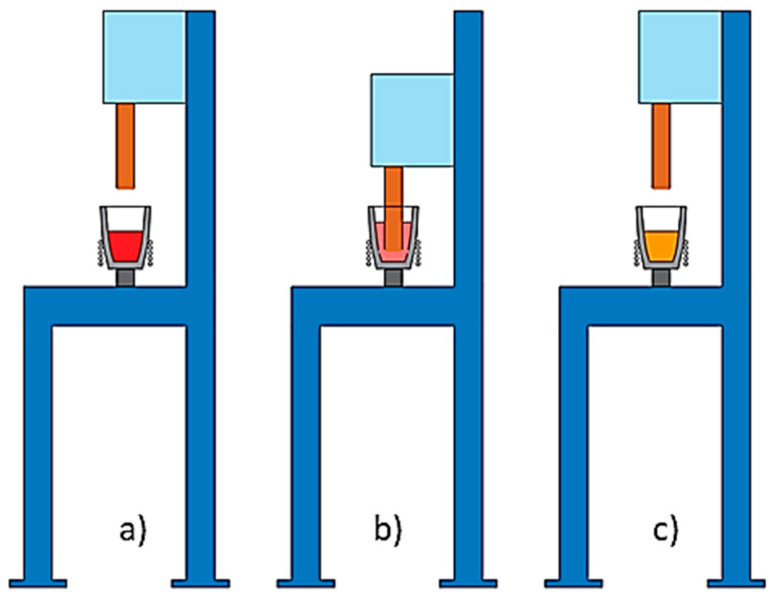
Steps of agitation process: (**a**) melting, (**b**) stirring, (**c**) post-processing (pouring).

**Figure 5 materials-16-02280-f005:**
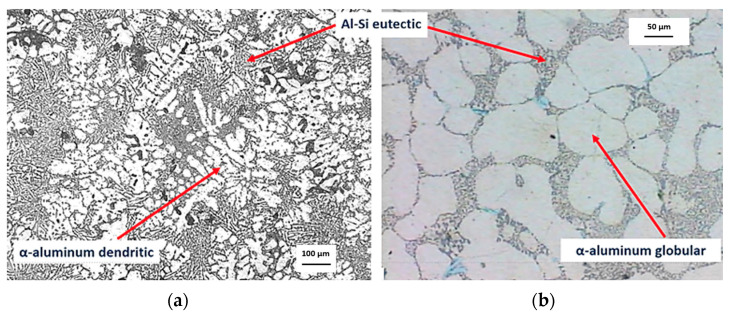
(**a**) α-aluminum dendritic microstructure of as-cast A356 aluminum alloy; (**b**) α-aluminum globular microstructure of the solidified SSM slurry (experiment carried out at a constant strain rate of 175 rpm).

**Figure 6 materials-16-02280-f006:**
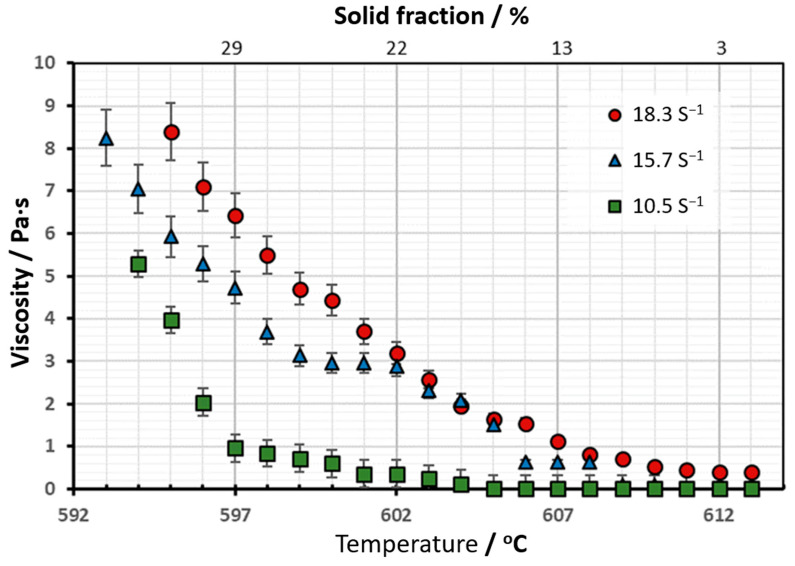
Evolution of viscosity vs. temperature and solid fraction: data obtained with the new rotation rheometer. Experiments carried out on the A356 aluminum alloy at a constant strain rate of 10.5 s^−1^ (100 rpm), 15.7 s^−1^ (150 rpm) and 18.3 s^−1^ (175 rpm).

**Figure 7 materials-16-02280-f007:**
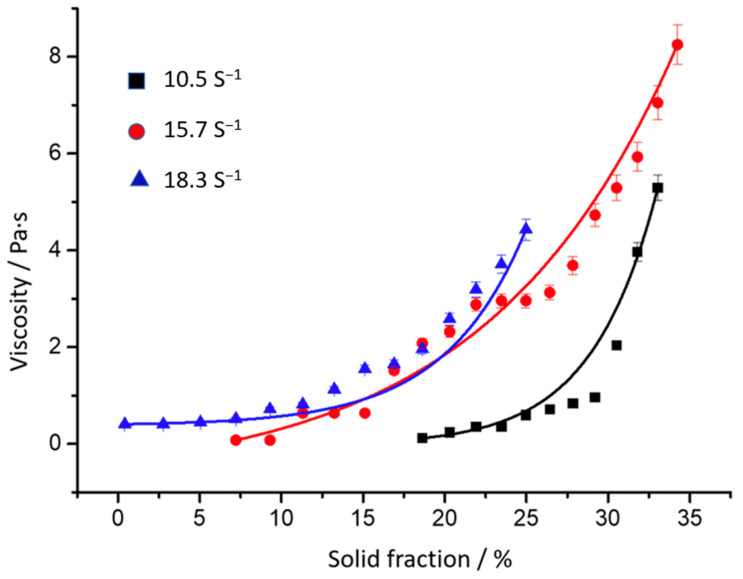
Evolution of viscosity vs. solid fraction.

**Figure 8 materials-16-02280-f008:**
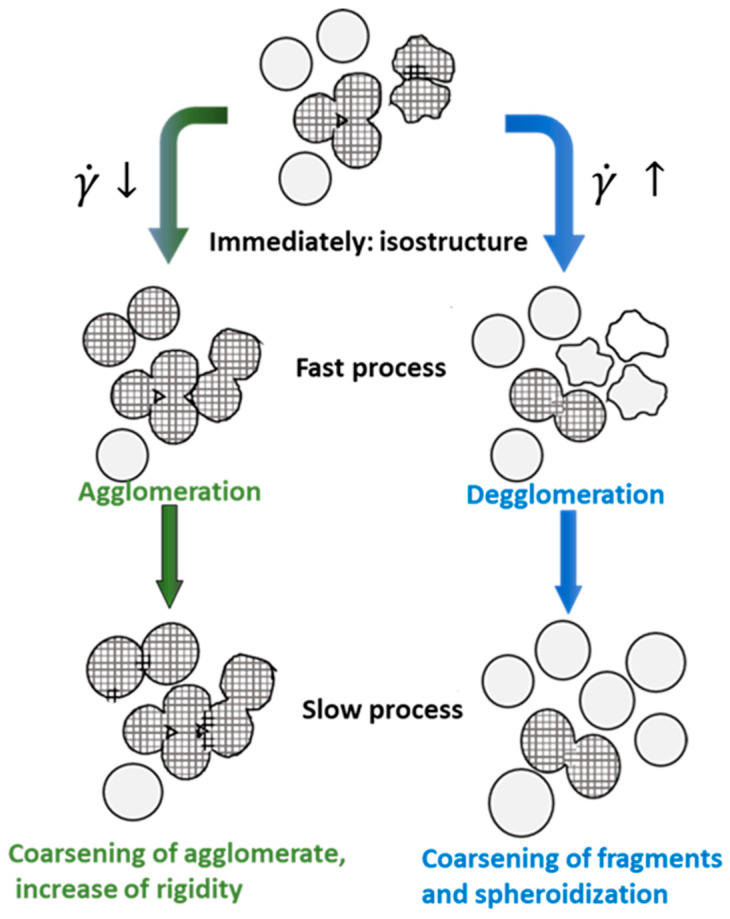
Qualitative model describing the two-step structure evolution after upwards and downwards shear rate jumps, adapted from [22].

**Table 1 materials-16-02280-t001:** Chemical composition of the used A356 aluminum alloy, wt %.

Si	Fe	Cu	Mn	Mg	Zn	Ti	Al
7.12	0.0218	0.0025	0.0049	0.335	0.1	0.104	Balance

**Table 2 materials-16-02280-t002:** Values of the coefficients A and B for the exponential law of relationship between η and *f_s_*.

Strain Rate /s^−1^	A /Pa·s	Standard Error of A	B	Standard Error of B	R^2^
10.5	0.0014	0.0031	0.2494	0.0678	0.9068
15.7	0.467	0.2278	0.0865	0.0131	0.9671
18.3	0.0257	0.0293	0.2025	0.0459	0.9005

**Table 3 materials-16-02280-t003:** Values of the coefficients *η_o_* and E for the Arrhenius law (Equation (2)).

Strain Rate / s^−1^	$\eta_{o}$ /Pa s	Standard Error of $\eta_{o}$	E /J·mol^−1^·K^−1^	Standard Error of E	R^2^
10.5	0.00005	0.00001	5.909	1.477	0.8931
15.7	0.0007	0.0001	1.938	0.158	0.9744
18.3	0.0011	0.0001	2.047	0.446	0.9101

## Data Availability

Data will be made available on request.

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
