# Peer review of "Rheological Behavior of the A356 Alloy in the Semisolid State at Low Shear Rates"

_materials, 2023, doi:10.3390/ma16062280_

Round 1
Reviewer 1 Report
The manuscript is written on a relevant topic of research but it can be improved after incorporating certain basic points in the text.
1. In the introduction the motivation is not well defined. In fact the reason for initiating this study needs to be very well emphasized.
2. As far as methodology is concerned the rheological study could be more elaborated in term of deformation and flow of materials exactly in case of the chosen material.
3. The author can improve the text by quoting self citation also and the previous work if carried out by them close to this study can be more quoted in the present context.
4. The manuscript seems to be quite ok in terms of English and grammar but there are areas of scopes to improve it further.. specially in the Introduction and conclusion section...
5. The application of the alloy in 3D -thixo printing could be well deciphered in the conclusion along with the future applications as well.
Author Response
We thank the reviewer for his/her positive evaluation of our manuscript. The specific remarks are addressed below.
- In the introduction the motivation is not well defined. In fact the reason for initiating this study needs to be very well emphasized.
Author reply: Dear reviewer, thank you very much for your valuable comment. Based on your suggestion, we have carefully revised the introduction emphasized the motivation for initiating this study
Our detailed responses to specific comments are given below, and revisions in the manuscript are highlighted in yellow.
The first part of the introduction has been updated in the revision as below:
The ASTM 52900:2015 standard is an attempt to regularize a new field of manufac-turing processes, Additive Manufacturing (AM), which has generated a lot of interest in diverse engineering sectors and has a great potential of expansion. This standard classi-fies the AM processes in the following categories: (1) Binder Jetting, (2) Directed Energy Deposition (DED), (3) Extrusion, (4) Material Jetting (5) Powder Bed Fusion (PBF), (6) Sheet Lamination and (7) VAT Photopolymerisation [1]. These technologies open new manu-facturing possibilities regarding the production of complex geometries, impossible to be produced by any other means, and the multi-functionality of the components. Despite the short time which has passed since the development of these technologies, their viability has been proven for different materials. Nowadays, the efforts are focused on the proper control of the processing parameters to assure the quality and properties of the printed components. Exploiting the whole potential of these technologies will require the devel-opment of new materials, and/or modifications of the existing ones, to optimize their rhe-ological behavior during printing. In general, the above-described technologies are the most extended techniques for the manufacturing of metallic parts by additive manufac-turing, but all of them have high costs associated to the heating systems, i.e. electron beam or laser.
- As far as methodology is concerned the rheological study could be more elaborated in term of deformation and flow of materials exactly in case of the chosen material.
Author reply: Dear reviewer, thank you very much for your valuable comment.
For the rheological study of the alloy in the semisolid state, a specific software was designed that compares the assigned torque/shear stress with the measured strain/strain rate. This information has been added to the manuscript.
In response to the reviewer's comment, the authors consider that the rheological study of metallic materials is a complex field, so we believe that the simplification of these concepts will facilitate the reader's understanding of the study developed.
- The author can improve the text by quoting self citation also and the previous work if carried out by them close to this study can be more quoted in the present context.
Author reply: Dear reviewer, thank you very much for your suggestion.
The following paragraph has been added to the introduction, adding some references to previous results obtained by the authors in the field of aluminum forming in the semisolid state:
The authors base this study on their extensive knowledge about the thixotropic forming of aluminum alloys [5-7, 9], whose rheological behavior in the semisolid state will be key for its printing by FDM type technologies.
References
5. Menargues, S.; Martin, E.; Baile, M.T.; Picas, J.A. New short T6 heat treatments for aluminium silicon alloys obtained by semisolid forming. Mater. Sci. Eng. A . 2015, 621, pp. 236-242. Doi: 10.1016/j.msea.2014.10.078.
6. Forn, A.; da Silva, M.; Baile, M.T.; Picas, J.A.; Fauria, A. Effect of ultrasounds during the solidification process of the A357 aluminum alloy. Solid State Phenom. 2013, 192-193, pp. 428-432. Doi: 10.4028/www.scientific.net/SSP.192-193.428.
7. Campillo, M.; Baile, M.T.; Menargues, S.; Martín, E.; Forn, A. A357 Aluminium Cast Alloys for Extrusion Processes. Solid State Phenom. 2013, 192-193, pp. 454-459. Doi: 10.4028/www.scientific.net/SSP.192-193.454
9. Menargues, S. Ph.D. Thesis. Optimización de componentes de aluminio obtenidos por Sub-Liquidus Casting. Universitat Politècnica de Catalunya, Vilanova i la Geltrú, 2011. http://hdl.handle.net/10803/6045.
- The manuscript seems to be quite ok in terms of English and grammar but there are areas of scopes to improve it further.. specially in the Introduction and conclusion section...
Author reply: We appreciate the valuable advice from the Reviewer. The English and grammar have been revised.
- The application of the alloy in 3D -thixo printing could be well deciphered in the conclusion along with the future applications as well.
Author reply: We would like to thank again the reviewer for the pertinent remarks. Based on your suggestion, we have modified some of the conclusions as follows:
5) After applying mechanical stirring in semisolid state, aluminum ingots can be used in a subsequent additive manufacturing process, based on the Fused Deposition Melting (FDM) techniques of metallic materials (3D thixo-printing).
6) The globulized alpha microstructure of aluminium ingots is considered beneficial, as a suitable microstructure to not impair the continuity of the semisolid alloy through the nozzle in the thixo-printing process. In this process, the material must be reheated to the semisolid state, in which the coexistence of a globular solid phase surrounded by a liquid phase, will allow the extrusion of the semisolid slurry with the appropriate viscosity. However, for very low shear rates, problems of possible clogging of the nozzles of additive manufacturing printers are foreseen, due to the dilating nature of the alloy.
7) The printing of metallic materials in semisolid state will introduce manufacturing possibilities never before foreseen. The reduction in the cost of printing will expand the use of metal printing to diverse industrial sectors.

Reviewer 2 Report
The manuscript “Rheological behavior of A356 alloy in semisolid state at low shear rates” presents modeling and controlling the semi-solid processing of aluminum alloys. Rheological properties of aluminum alloy were carried out for the low sheer rates (three selected values from 10.5 1/s to 18.3 1/s). The authors prepared a well-written manuscript. All results are clearly discussed and presented. Unfortunately, a potential reader may feel a bit unsatisfied after reading the manuscript. The introduction is a bit general and includes four figures from other articles. Moreover, the purpose of their inclusion is not always clearly defined. In addition, the data presented in Fig. 10 (experimental viscosity values) and Fig. 11 (theoretical fits) should be summarized in one figure, etc. However, the manuscript is very interesting, and in my opinion, it should be published in the Journal after major amendments.
1. Page 2, line 61 – Eq. (1): Add the meanings of the coefficients “A” (pre-exponential factor) and “B”, and “fs” (solid fraction).
2. Page 2, lines 72-74: The unit for the flow consistency index “m” should be added ([Pa s^n]), and information for values of the index “n”, i.e. for n<1 pseudoplastic fluid, n = 1 Newtonian fluid, n>1 dilatant fluid (or shear-thickening fluid).
3. Figure 1: Give the meninges of the numbers 1-7. Give the meaning of the dashed curve in Fig. 1. Give the meaning of the “m.p.” shortcut used (melting point).
4. Figures 2, 3, 4, 9: Add the temperature of the measurements.
5. General question and remarks: Why did the authors show Figs. 1-4? Discussion of the properties shown in these figures is feeble and, I think, it brings no crucial information concerning the scientific meaning of the new data and results present in the manuscript. Please, consider the need to include these figures (taken from other articles). A potential reader can find these figures by themselves. The results are only important! In addition, the quality of Fig. 4 is very poor!
6. Page 7, line 171: What is the meaning of “L”? It is crucial information for a potential reader! Is it the same as in Eq. (6)? What is the meaning of “alpha”? Please, be careful when you give shortcuts, etc. All of them should be defined after the first use, and the authors have to avoid any conflict of abbreviations/shortcuts used.
7. General error: There are commas instead of points between the integer and fraction parts – see page 7, line 199; page 8, line 217; page 9, line 229 (figure caption); page 9, Fig. 11; page 11, line 301.
8. Figure 10: What is the error of the viscosity measurements? Specify it! Why did the measurement for the shear rate of 18.3 1/s start from 600 degrees?
9. Page 9, lines 231-237: I think the authors should check the results described there with the data presented in Fig. 10 for the lower shear rate (i.e. 105 1/s). For me and, I hope so, for a potential reader, no appreciable difference in apparent viscosity is noticed even up to 19% (the authors gave 15%). The (slow/medium) raised of the viscosity is seen between 19% and 30 % (the authors gave between 15% and 20%). The fast raised of the viscosity values is noticed for the fraction higher than 30% (the authors gave 20%)!
10. Page 9, lines 238-240, Figs. 10 and 11: Show the fit results in Fig. 10, instead of joined experimental points. Figure 11 is out of the sense. The results fitted should be shown with the experimental data (together!!!).
11. Table 2: The coefficient “A” should have the unit. Add errors for the coefficients “A” and “B”.
12. Table 3: Add errors for the coefficients “eta_0” and “E”.
13. Page 11, lines 307-308: The authors wrote: “When the solid fraction exceeded 25%, it was observed that viscosity increased rapidly.” (compare with the results present on page 9, lines 234-235: “(…) when the solid fraction exceeded 20%, it was observed that viscosity increased rapidly”). Please, see my comment – point 9! Please be consistent in the results presented!
14. Minor errors found: (i) page 2 line 65 – Eq. (2): There is “K” in the denominator, and it should be “R” (the gas constant); (ii) page 5, line 120 and line 127: meanings of “tau” (the shear stress) and “gamma-point” (the shear rate) are given on page 2, lines 56 and 121. There is no need to repeat this information; (iii) page 5, lines 121 and 127: “R_i” and “R_a” – the small case letters “i” and “a” should be subscripts; (iv) page 5, lines 118 and 120: “T” is the temperature (see Eq. (2)), so to avoid an abbreviation conflict, consider to use another symbol for a torque (for example bolded “T”); (v) the authors use “experimentation” and I hope it will be better to use “experiment” (page 5, line 135; page 6, line 161); (vi) page 5, lines 135-137: please specify that the rotor with a diameter 50 mm is “R_i”, and the cup with a diameter 110 mm is “R_a”.
Author Response
Comments and Suggestions for Authors
The manuscript “Rheological behavior of A356 alloy in semisolid state at low shear rates” presents modeling and controlling the semi-solid processing of aluminum alloys. Rheological properties of aluminum alloy were carried out for the low sheer rates (three selected values from 10.5 1/s to 18.3 1/s). The authors prepared a well-written manuscript. All results are clearly discussed and presented. Unfortunately, a potential reader may feel a bit unsatisfied after reading the manuscript. The introduction is a bit general and includes four figures from other articles. Moreover, the purpose of their inclusion is not always clearly defined. In addition, the data presented in Fig. 10 (experimental viscosity values) and Fig. 11 (theoretical fits) should be summarized in one figure, etc. However, the manuscript is very interesting, and in my opinion, it should be published in the Journal after major amendments.
We thank the reviewer for his/her positive evaluation of our manuscript. The specific remarks are addressed below.
- Page 2, line 61 – Eq. (1): Add the meanings of the coefficients “A” (pre-exponential factor) and “B”, and “fs” (solid fraction).
Author reply: Dear reviewer, thank you very much for your valuable comment. Based on your suggestion, we have added the information in the paragraph as follows:
where A is related to the pre-exponential viscosity, B to the activation energy, and fs is the solid fraction. Accordingly, equation 1 can be considered as a form of the expression of the Arrhenius equation [16]:
- Page 2, lines 72-74: The unit for the flow consistency index “m” should be added ([Pa s^n]), and information for values of the index “n”, i.e. for n<1 pseudoplastic fluid, n = 1 Newtonian fluid, n>1 dilatant fluid (or shear-thickening fluid).
Author reply: We appreciate the valuable advice from the Reviewer. Based on your suggestion, we have added the information in the paragraph as follows:
where m is the flow consistency index (Pa·sn), and n is the flow behaviour index (dimen-sionless). Power-law fluids can be subdivided into three different types of fluids based on the value of their flow behaviour index. For n < 1 pseudoplastic fluid, n = 1 Newtonian fluid, n > 1 dilatant fluid (or shear-thickening fluid).
- Figure 1: Give the meninges of the numbers 1-7. Give the meaning of the dashed curve in Fig. 1. Give the meaning of the “m.p.” shortcut used (melting point).
- Figures 2, 3, 4, 9: Add the temperature of the measurements.
- General question and remarks: Why did the authors show Figs. 1-4? Discussion of the properties shown in these figures is feeble and, I think, it brings no crucial information concerning the scientific meaning of the new data and results present in the manuscript. Please, consider the need to include these figures (taken from other articles). A potential reader can find these figures by themselves. The results are only important! In addition, the quality of Fig. 4 is very poor!
Author reply: Dear reviewer, thank you very much for your valuable comment. Based on your useful suggestions, we have removed graphs 1-4 from the manuscript. We agree that these graphs do not provide crucial information about the scientific significance of the data and results of this manuscript.
- Page 7, line 171: What is the meaning of “L”? It is crucial information for a potential reader! Is it the same as in Eq. (6)? What is the meaning of “alpha”? Please, be careful when you give shortcuts, etc. All of them should be defined after the first use, and the authors have to avoid any conflict of abbreviations/shortcuts used.
Author reply: Dear reviewer, thank you very much for your valuable remark. We have added the information in the paragraph as follows: liquid + solid phase of α-aluminum
- General error: There are commas instead of points between the integer and fraction parts – see page 7, line 199; page 8, line 217; page 9, line 229 (figure caption); page 9, Fig. 11; page 11, line 301.
Author reply: Dear reviewer, thank you very much for your valuable observation. These grammatical errors have been corrected.
- Figure 10: What is the error of the viscosity measurements? Specify it! Why did the measurement for the shear rate of 18.3 1/s start from 600 degrees?
Author reply: Dear reviewer, thank you very much for your valuable comment. The error of the viscosity measurements has been added to Figure 10 (Figure 6, after review). In addition, we thank the reviewer to pointing out the missing part of the curve (Figure 6), corresponding to the test carried out at 18.3 1/s, which has now been included.
- Page 9, lines 231-237: I think the authors should check the results described there with the data presented in Fig. 10 for the lower shear rate (i.e. 105 1/s). For me and, I hope so, for a potential reader, no appreciable difference in apparent viscosity is noticed even up to 19% (the authors gave 15%). The (slow/medium) raised of the viscosity is seen between 19% and 30 % (the authors gave between 15% and 20%). The fast raised of the viscosity values is noticed for the fraction higher than 30% (the authors gave 20%)!
- Page 9, lines 238-240, Figs. 10 and 11: Show the fit results in Fig. 10, instead of joined experimental points. Figure 11 is out of the sense. The results fitted should be shown with the experimental data (together!!!).
Author reply: The authors agree with the reviewer's comment. We have checked and modified accordingly the data presented in Figure 6 (Figure 10, before review). Figure 7 (Figure 11, before review) shows the evolution of the viscosity with the solid fraction from the values calculated in table 2.
- Table 2: The coefficient “A” should have the unit. Add errors for the coefficients “A” and “B”.
- Table 3: Add errors for the coefficients “eta_0” and “E”.
Author reply: Dear reviewer, thank you very much for your valuable comment.
In tables 2 and 3 the square of the correlation (R2) has been added, which is a useful value in linear regression. This value represents the fraction of the variation in one variable that may be explained by the other variable. A value of 1 indicates a perfect correlation.
- Page 11, lines 307-308: The authors wrote: “When the solid fraction exceeded 25%, it was observed that viscosity increased rapidly.” (compare with the results present on page 9, lines 234-235: “(…) when the solid fraction exceeded 20%, it was observed that viscosity increased rapidly”). Please, see my comment – point 9! Please be consistent in the results presented!
Author reply: Dear reviewer, thank you very much for your valuable observation. These values have been corrected.
- Minor errors found: (i) page 2 line 65 – Eq. (2): There is “K” in the denominator, and it should be “R” (the gas constant);
(ii) page 5, line 120 and line 127: meanings of “tau” (the shear stress) and “gamma-point” (the shear rate) are given on page 2, lines 56 and 121. There is no need to repeat this information;
(iii) page 5, lines 121 and 127: “R_i” and “R_a” – the small case letters “i” and “a” should be subscripts;
(iv) page 5, lines 118 and 120: “T” is the temperature (see Eq. (2)), so to avoid an abbreviation conflict, consider to use another symbol for a torque (for example bolded “T”);
(v) the authors use “experimentation” and I hope it will be better to use “experiment” (page 5, line 135; page 6, line 161);
(vi) page 5, lines 135-137: please specify that the rotor with a diameter 50 mm is “R_i”, and the cup with a diameter 110 mm is “R_a”.
Author reply: The authors thank the reviewer for highlighting these errors. In accordance with the reviewer’s suggestions, the text has been corrected and modified.

Round 2
Reviewer 2 Report
The manuscript “Rheological behavior of A356 alloy in semisolid state at low shear rates” has been amended with (almost) all of my suggestions/doubts and the authors answer all my questions. However, I cannot agree with the authors’ answers to point1 11 and 12 – concerning the need to add the errors of estimated parameters “A” and “B”, “eta_0” and “E”. I think that the errors of the estimated/calculated parameters should be explicitly given or, at least, the percentage of the errors (with respect to the values obtained) determined. Please, see that the R-square parameters are rather low (much lower than 0.9!!!) for the shear rates of 10.5 1/s and 15.7 1/s, which means that the fitted theoretical curves lie away from some experimental points. In the other words, the errors of “A” and “B”, “eta_0” and “E” parameters are high! I also find some very minor misprints (given below):
(1) Page 7, line 248: There is a point after the bracket (no need for the point, a comma could be there).
(2) Page 4, line 155; page 7, line 249; page 9, line 290: There are “fs” and the small case letter “s” should be as a subscript.
I think that after amendments and mentioning the errors of calculated parameters, the potential reader will be very pleased and fascinated by the article. Thus, I recommend publishing the manuscript after minor amendments.
Author Response
Response to Reviewer 2 Comments
We thank again the reviewer for his/her positive evaluation of our manuscript.
Our detailed responses to specific comments are given below and revisions in the manuscript are highlighted in yellow.
We have carefully reviewed the calculations made of parameters “A” and “B”, “eta_0” and “E” (Tables 2 and 3), updating these values and adding the standard deviation. We have also updated Figure 7 according to the new values.
The other minor errors in the text have been corrected.